# Optimizing CMV therapy: Population pharmacokinetics and Monte Carlo simulations for letermovir and maribavir dosage

Yeleen Fromage[1], Hamza Sayadi[1], Sophie Alain[2,3,4,5], Pierre Marquet[1,5,6], Gilles Peytavin[7,8], Jean-Baptiste Woillard[1,5,6]*

**1** Service of Pharmacology, Toxicology and Pharmacovigilance, CHU Limoges, Limoges, France, **2** Microbiology Department, CHU Limoges, Limoges, France, **3** RESINFIT, UMR1092, Inserm, Univ. Limoges, CHU Limoges, Limoges, France, **4** National Reference Center for Herpesviruses, CHU Limoges, Limoges, France, **5** Fédération Hospitalo-Universitaire Survival Optimization in Organ Transplantation (FHU SUPORT), Limoges, France, **6** Inserm UMR1248, Pharmacology & Transplantation, Univ. Limoges, Limoges, France, **7** AP-HP Nord, Pharmacology Department, Bichat Claude-Bernard University Hospital, Paris, France, **8** Inserm, UMR 1137, Univ. Paris Cité, Paris, France

* jean-baptiste.woillard@unilim.fr

## Abstract

### Purpose

This study seeks to reassess and enhance the dosing strategies of letermovir and maribavir for treating cytomegalovirus (CMV) infection, aiming to propose adjustments that could improve therapeutic effectiveness.

### Methods

Pre-existing population pharmacokinetic models were used alongside Monte Carlo simulations to evaluate the dosing strategies of letermovir and maribavir in CMV treatment. The simulations assessed the probability of target attainment for current and alternative dosing regimens, including scenarios with missed doses.

### Results

For letermovir, a loading dose on the first day of treatment initiation appeared more effective than the current strategy without a loading dose. Additionally, in cases of missed doses, doubling the dose upon resumption was more effective than returning to the normal dosage. For maribavir, the current 400mg BID regimen only covers the lower end of the inhibitory concentration 50 range, suggesting a potential benefit from increasing the doses. Simulations indicated that for missed doses, all tested regimens only covered the lower range of inhibitory concentrations, but the current strategy of resuming the normal dosage provided the lowest chances of target attainment.

**Data availability statement:** All relevant data are within the paper and its Supporting Information files.

**Funding:** SA received research grants or contracts paid to institutions, travel grants, advisory board fees, and speaker fees from MSD and Takeda. GP received travel grants and speaker fees from MSD and Takeda, and PM received speaker fees from Takeda. These companies have no role in the study design, data collection and analysis, decision to publish, or preparation of the manuscript.

**Competing interests:** SA has received research grants or contracts paid to institutions, travel grants, advisory board fees, and speaker fees from MSD and Takeda. GP has received travel grants and speaker fees from MSD and Takeda, and PM has received speaker fees from Takeda. However, none of the authors are employed by these companies. This does not alter our adherence to PLOS ONE policies on sharing data and materials.

## Conclusion

Our findings suggest a strong rationale to reconsider and potentially modify the approved dosing guidelines for letermovir and maribavir in CMV treatment. Adjusting dosing regimens, including the use of loading doses and increased doses after missed doses, could enhance treatment outcomes by ensuring higher probabilities of achieving therapeutic targets and better managing missed doses.

## Introduction

Most anti-cytomegalovirus (CMV) agents act through the inhibition of CMV deoxyribonucleic acid polymerase (DNA), with resistance mechanisms emerging due to mutations or deletions in the UL97 or UL54 CMV DNA polymerase [1,2]. One of the risk factors for resistance emergence is prolonged exposure to anti-CMV drugs or prior exposure to an anti-CMV agent [3]. This is particularly problematic because anti-CMV prophylaxis is frequent and prolonged in the context of transplantation. These findings have led to the development of new non-nucleoside anti-CMV medications, including letermovir and maribavir.

Letermovir, a direct inhibitor of the CMV terminase complex, exhibits no cross-resistance with other anti-CMV drugs and has the lowest inhibitory concentrations among anti-CMV agents but has a low genetic barrier [4]. It is available in film-coated tablets or for intravenous administration as a diluted solution for infusion at doses of 240 or 480 mg [5]. Maribavir, on the other hand, is only available orally in 200 mg tablets and acts on the CMV replication cycle by competing with ATP for binding to the UL97 protein kinase [6,7]. As a consequence, it inhibits nuclear egress and, to a lesser extent, DNA replication and encapsidation.

Another potential factor for resistance emergence can be inadequate dosing regimens of anti-CMV drugs, leading to ineffective suboptimal plasma concentrations. This is particularly true for letermovir due to its mechanism of action with plasma concentration directly linked to its efficacy. Saturating CMV-infected cells with medication as quickly as possible, beginning from the first days of treatment initiation before reaching steady state, in order to reduce viral load as rapidly as possible could thus be envisaged. For maribavir, as its mechanism of action is more indirect, the benefit of initiating treatment with loading doses is less clear. However, increasing overall drug exposure by raising the dosage is an interesting avenue to explore. Moreover, the failure of low dose to prevent CMV infection in prophylaxis studies was partly explained by low plasma concentrations [8]. Indeed, letermovir has demonstrated approximately a 20% virological failure rate in prophylaxis studies [9], while maribavir has shown around a 45% failure rate in therapeutic studies in difficult-to treat patients [10]. In this context, therapeutic drug monitoring (TDM) can be useful and interesting to detect suboptimal exposure.

Various pharmacodynamic markers are used to link measured pharmacokinetic exposure to antiviral efficacy. The *in vitro* 50% and 90% inhibitory concentrations (IC50 and IC90) are the extracellular concentrations able to inhibit 50% or 90% of

viral growth, respectively. This relies on *in vitro* assays in cell-culture with either clinical isolates or recombinant viruses. Using IC50–90 as a reference suggests that low concentrations may not completely inhibit the viral replication. This is particularly true for direct inhibitors that do not accumulate as an activated triphosphate nucleotide in the infected cell as for ganciclovir [11]. Letermovir reported IC50 values in the literature range from 0.00086 mg/L to 0.00166 mg/L [5,12] and maribavir reported IC50 values in the literature range from 0.0452 mg/L to 0.2107 mg/L [7]. These IC50 values were obtained from a collection of clinical CMV isolates.

Population pharmacokinetics (POPPK) enables the characterization of the path of a drug following its administration, quantifying and identifying its sources of variability, and facilitating simulations. Through simulations, various hypothetical scenarios can be assessed [13,14]. It is possible to evaluate different "what if" scenarios, especially consequence of increase the dosage through loading doses as well as missing dose on drug exposure.

The objective of this work was i) to implement population pharmacokinetic models from literature for letermovir and maribavir, ii) to simulate different dosing administration regimens using Monte Carlo simulations, more specifically the impact of loading doses for letermovir and dosage escalation for maribavir on plasma concentrations, iii) to simulate missed doses and therapeutic regimens for resumption to highlight the risk of resistance emergence in the case of missed doses.

## Materials and methods

This study was approved by the ethics committee with approval number 03-2025-01.

### Population pharmacokinetic models from literature

**Letermovir.** The letermovir POPPK model [15] was developed from 280 healthy participants enrolled in phase 1 trial and from 399 HSCT (hematopoietic stem-cell transplantation) recipients enrolled in phase 2 and 3 trials [9,16,17]. Briefly, it consisted in a two-compartment structural model with first-order lag-time absorption and linear elimination from the central compartment. In this model, cyclosporine (CsA) co-treatment was identified as a covariate on clearance and bioavailability (in the original article, a covariate healthy vs CMV patient was described but we only draw simulations for CMV patients). Parameters of the final model reported in the original publication were used with a few modifications: the proportional and additive errors were decreased to 0.0001 in order to only account for the inter-individual variability and covariate effects as previously described [18]. The PK parameters of the models implemented are reported in the Table 1.

**Maribavir.** For maribavir the POPPK model [19] consisted in a two-compartment model with first-order lag-time absorption and linear elimination developed from participants enrolled in nine phase 1 studies, two phase 2 studies and one phase 3 study [10,20]. In this model, body weight was identified as a covariate on clearance and volume parameters. The PK parameters implemented are reported in Table 1. Note that in this model, there was no additive error, and likewise, the proportional error was reduced to 0.0001 to only account for inter-individual variability and covariate effect.

**Table 1. Typical pharmacokinetic parameters used for simulations[a].**

| Drugs | Cl/F (L.h⁻¹) | Vc/F (L) | Vp/F (L) | Ka (h⁻¹) | Q (L/h) | Alag (h) | Original ε add | Original ε prop | ε add used | ε prop used |
|---|---|---|---|---|---|---|---|---|---|---|
| Letermovir | 12.5 | 19.7 | 25.8 | 0.15 | 1.54 | 0.674 | 0.383 | 0.517 | 0.0001 | 0.0001 |
| Maribavir | 3.9 | 19 | 8.5 | 0.3 | 0.83 | 0.26 | NA | 0.0671 | 0 | 0.0001 |

[a]Abbreviations: Cl/F mean apparent clearance; Vc/F mean apparent central volume of distribution; Vp/F mean apparent peripherical volume of distribution; Ka mean absorption rate, Q mean intercompartimental clearance, Alag mean lag absorption time; $\varepsilon_{add}$ mean additive residual error; $\varepsilon_{prop}$ mean proportional residual error, NA: Not available: no additional error in this study

## Simulations

In this work, 'normal doses' refers to the doses specified in the Summary of Product Characteristics (SPC) of the drugs leading to their Marketing Authorization (MA): 480 mg/day and 240 mg/day administered orally or as an intravenous infusion over 1 hour respectively for letermovir without and with co-treatment by CsA [5], and 400 mg BID administered orally for maribavir [7]. A 'loading dose' of letermovir refers to doubling the standard dosage (i.e., 860 mg for letermovir without CsA and 480 mg with it). POPPK models from the literature were implemented for each drugs in the mrgsolve R package [21].

Currently, there is no consensus on specific PK/PD targets to ensure therapeutic effectiveness. Consequently, we chose to compare our simulated plasma concentrations to the PA-IC50 (Pharmacologically Active Inhibitory Concentration 50%) values reported in the literature for letermovir and maribavir, as well as to the commonly used threshold of $5*$PA-IC50. We performed probability of target attainment (PTA) analyses based on these inhibitory concentration indices, utilizing 10,000 simulations for each of the scenarios outlined below. Notably, this study was conducted using wild-type phenotypes that did not exhibit any signature resistance mutations.

**Letermovir.** For letermovir, 10,000 Monte Carlo simulations for typical doses without and with CsA were drawn for several scenarii: i) without loading dose; ii) with a loading dose on Day 1; iii) with a loading dose on days 1 and 2; iv) with a loading dose from day 1 to day 3 of treatment initiation. Letermovir trough plasma concentration (C24h) were simulated at each day from day 1 to day 10 in order to study the impact of these different scenarii in terms of rate of plasma concentration at steady state (Css) attainment. Considering a plasma protein binding rate of 98.2% [5], the target concentration intervals for PA-IC50 are [0.047–0.092] mg/L, and [0.235–0.460] mg/L for $5*$PA-IC50. This study was conducted for oral administration and intravenous administration (with a one-hour infusion time).

To find the best dosing plan after a missed dose, we examined three scenarii for the next dose: (i) resuming at the usual dosage, (ii) doubling the once-daily dosage, and (iii) splitting the usual daily dose in twice daily. We then simulated two types of C24h: (i) at 48 hours after the last dose (to understand the missed dose's impact) and (ii) at 24 hours after resuming treatment (considering the three scenarii mentioned). This missed dose study only considered oral administration. Intravenous infusions, typically administered in hospitals, have a lower risk of missed doses. Both the loading dose and missed dose simulations were run with and without CsA co-administration.

**Maribavir.** For maribavir, 10,000 Monte Carlo simulations were drawn for five scenarii: i) 400 mg/12h (normal oral twice-daily dosage); and four scenarii of dosage escalations ii) 400 mg/8h; iii) 600 mg/12h; iv) 800 mg/12h and v) 800 mg/8h. Plasma maximum concentrations (Cmax) and trough concentrations (pre-dose) were simulated according to these scenarii to study these different dosing regimens. The body weight covariate was simulated based on truncated normal distribution following the values from a normal-weight population: weight = mean±SD [min-max] = $75 \pm 10$ [50–120] kg in accordance to the original article [19]. Considering a plasma protein binding rate of 98% [7], the target concentration intervals for PA-IC50 are [2.56–10.53] mg/L, and [11.29–52.67] mg/L for $5*$PA-IC50.

Regarding missed doses, there were some differences with letermovir due to the maribavir's formulation (200 mg tablets) and its approved dosage: 400 mg every 12 hours. Thus, C24h were simulated at two time points: i) at T = 12h after the last dose to study the impact of a missed dose, ii) at T = 12h after resuming treatment according to different rescue dosing scenarii. These scenarii include: i) standard dosage of 400 mg, ii) 1.5 times the standard dosage, equivalent to 600 mg facilitated by the 200 mg tablet formulation, and iii) double dosage, i.e., 800 mg.

## Statistical analyses

The reliability of the model implementation was assessed based on the comparison of the median C24h with the ones observed in the literature for normal doses. Histograms of simulated C24h were drawn and filters based on percentiles were applied to remove extreme simulated values (outliers) [22]. Four percentile filters were investigated and compared with the simulated C24h as a sensitivity analysis: ([quantile_1-quantile_99], [quantile_2.5-quantile_97.5], [quantile_5-quantile_95] and [quantile_10-quantile_90]).

## Results

### Simulations with letermovir

The histogram of simulated letermovir C24h are shown in S1 Fig.

Letermovir's median (IQR) C24h values (mg/L) showed little variation across all quartile ranges ([1–99], [2.5–97.5], [5–95], and [10–90]) in Table 2. This indicates that the median C24h remained consistent regardless of the specific percentile analyzed.

### Simulations with maribavir

Similarly, maribavir's median (IQR) C24h values across all quantile ranges ([1–99], [2.5–97.5], [5–95], and [10–90]) were similar as shown in Table 2. For example, the median (IQR) C24h in the 1–99 percentile range was 7.8 (4.9–11.9) mg/L.

### Study of the different dosing regimens

**Letermovir.** S1 and S2 Tables show simulated letermovir concentrations (C24h) from days 1–10 for various loading dose regimens. When administered orally without a loading dose (MA regimen), steady-state (SS) appears to be reached by day 5 or 6. This means C24h stabilize by day 10. More precisely, the median (IQR) C24h is 1.28 (1.27–1.29) mg/L at day 6 with cyclosporine and 0.588 (0.526–0.659) mg/L at day 5 without cyclosporine. Loading doses ensure that C24h never falls below these SS values. Fig 1A presents the PTA after the initial administration of letermovir, comparing the standard dosage (480 mg without a loading dose) to a regimen with a loading dose (960 mg). This comparison is made in scenarii when the drug is not co-administered with cyclosporine.

For intravenous administration without a loading dose (MA regimen), SS is reached on day 5 with cyclosporine and day 4 without cyclosporine. Similar to the oral route, C24h remains stable thereafter. Specifically, the median (IQR) C24h is 1.31 (1.29–1.32) mg/L at day 5 with cyclosporine and 0.591 (0.526–0.663) mg/L at day 4 without cyclosporine. Loading doses prevent C24h from dipping below these SS levels. Fig 2 illustrates typical letermovir simulations for different loading doses. All these simulated concentrations remained higher than the target concentrations for PA-IC50 ([0.047–0.092] mg/L), and for 5*PA-IC50 ([0.235–0.460] mg/L).

**Maribavir.** Regarding maribavir, the simulated Cmax and trough plasma concentrations for the different dosing regimens are summarized in Table 3. Both Cmax and C12h/C8h increase linearly with increasing dosage up to 800 mg

**Table 2. Results of simulations on trough plasma concentrations depending on different quantile of extreme patients filtered out for letermovir and maribavir.**

| Anti-CMV | Concentrations (median [IQR]) | Quantile [10–90] | Quantile [5–95] | Quantile [2.5–97.5] | Quantile [1–99] |
|---|---|---|---|---|---|
| LETERMOVIR DOSE = 480mg/Day | Trough concentration at 24h (mg/L) | 0.57 | 0.57 | 0.57 | 0.57 |
| | | [0.52–0.62] | [0.51–0.63] | [0.51–0.64] | [0.51–0.64] |
| LETERMOVIR DOSE = 240mg/Day* | Trough concentration at 24h (mg/L) | 1.24 | 1.24 | 1.24 | 1.24 |
| | | [1.23–1.25] | [1.23–1.25] | [1.23–1.25] | [1.23–1.25] |
| MARIBAVIR DOSE = 400 mg BID | Trough concentration at 12h (mg/L) | 7.79 | 7.79 | 7.78 | 7.79 |
| | | [5.41–10.95] | [5.13–11.50] | [4.80–11.77] | [4.88–11.94] |

*with cyclosporine

Target concentrations

• letermovir for PA-IC50 ([0.047–0.092] mg/L), and for 5*PA-IC50 ([0.235–0.460] mg/L)

• maribavir for PA-IC50 ([2.56–10.53] mg/L), and for 5*PA-IC50 ([11.29–52.67] mg/L)

For instance, at a dosage of 480mg/day, our study found a median (IQR) C24h of 0.57 (0.501–0.64) mg/L. For the 240mg/day dosage with cyclosporine co-administration, the median (IQR) C24h was 1.24 (1.23–1.25) mg/L.

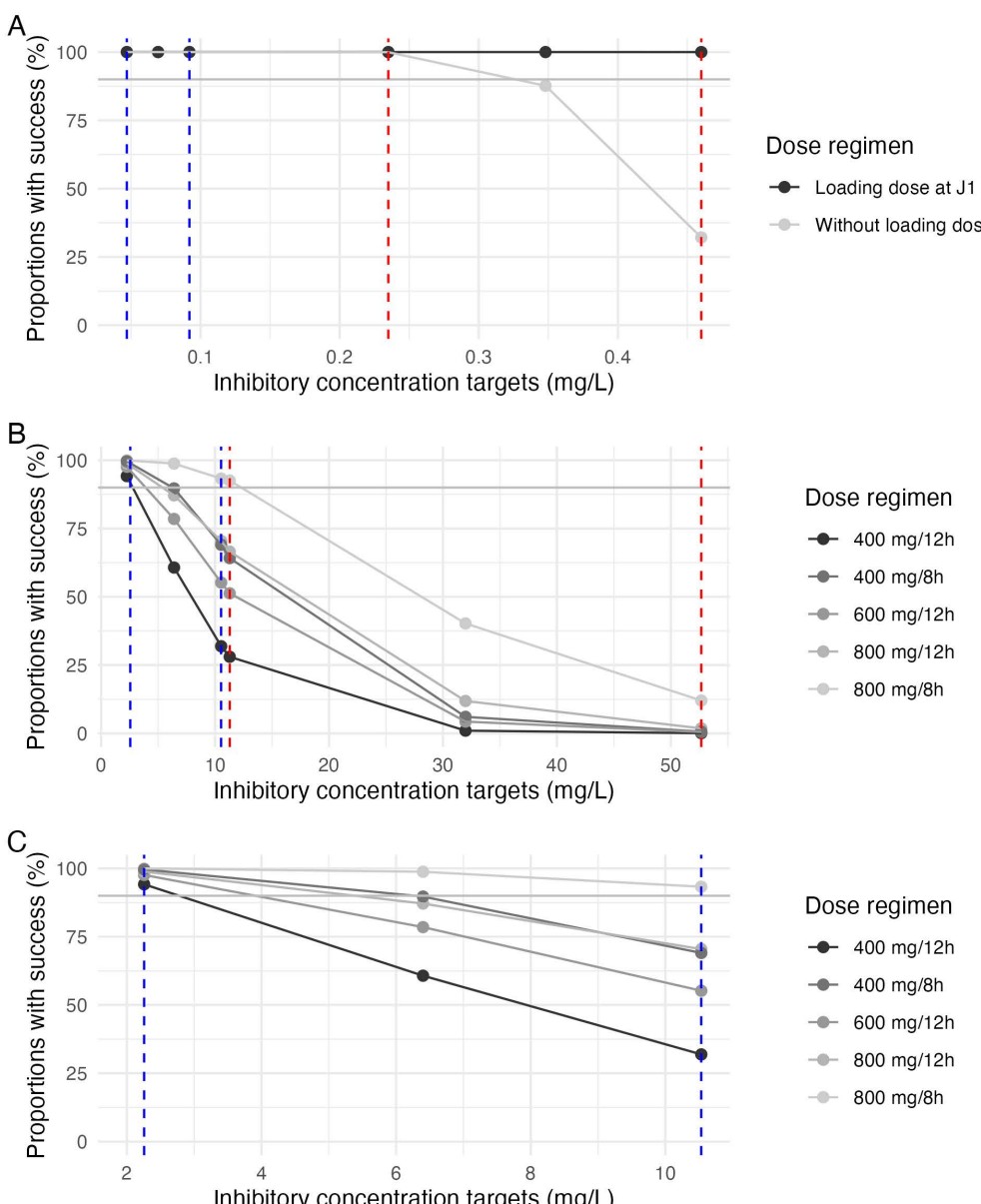

**Fig 1. Probability of target attainment (PTA) for different trough plasma concentration higher than inhibitory concentration or 5 times PA-IC50 across different dosing regimens for letermovir (A) and maribavir (B and C).** Abbreviations: **A:** PTA for letermovir administered without co-treatment by CsA across the range of PA-IC50 to 5*PA-IC50, with a dose of 480 mg without a loading dose and 960 mg with a loading dose; **B:** PTA for maribavir across the range of PA-IC50 to 5*PA-IC50; **C:** PTA for maribavir across the range of PA-IC50. The blue dashed lines represent the PA-IC50 ranges, specifically [0.047–0.092] mg/L for letermovir and [2.560–10.530] mg/L for maribavir. The red dashed lines represent the 5*PA-IC50 ranges, specifically [0.235–0.460] mg/L for letermovir and [11.290–52.670] mg/L for maribavir. The grey solid line represents the proportion of 90% success. Recommended dose regimens are 480 mg/day and 240 mg/day administered orally or as an intravenous infusion over 1 hour respectively for letermovir without and with co-treatment by CsA, and 400 mg BID administered orally for maribavir.

TID. Specifically, the median (IQR) Cmax and C24h respectively increase from 14.9 (11.3–19.7) mg/L and 7.7 (4.7–11.8) mg/L for the standard dosage of 400 mg BID to 39.2 (29.5–52.6) mg/L and 27.9 (18.8–40.3) mg/L for 800 mg TID. Fig 1B and 1C illustrate the PTA in terms of both PA-IC50 and 5*PA-IC50 for the various dosing regimens tested. As a reminder, the target concentrations for PA-IC50 are [2.56–10.53] mg/L, and [11.29–52.67] mg/L for 5*PA-IC50.

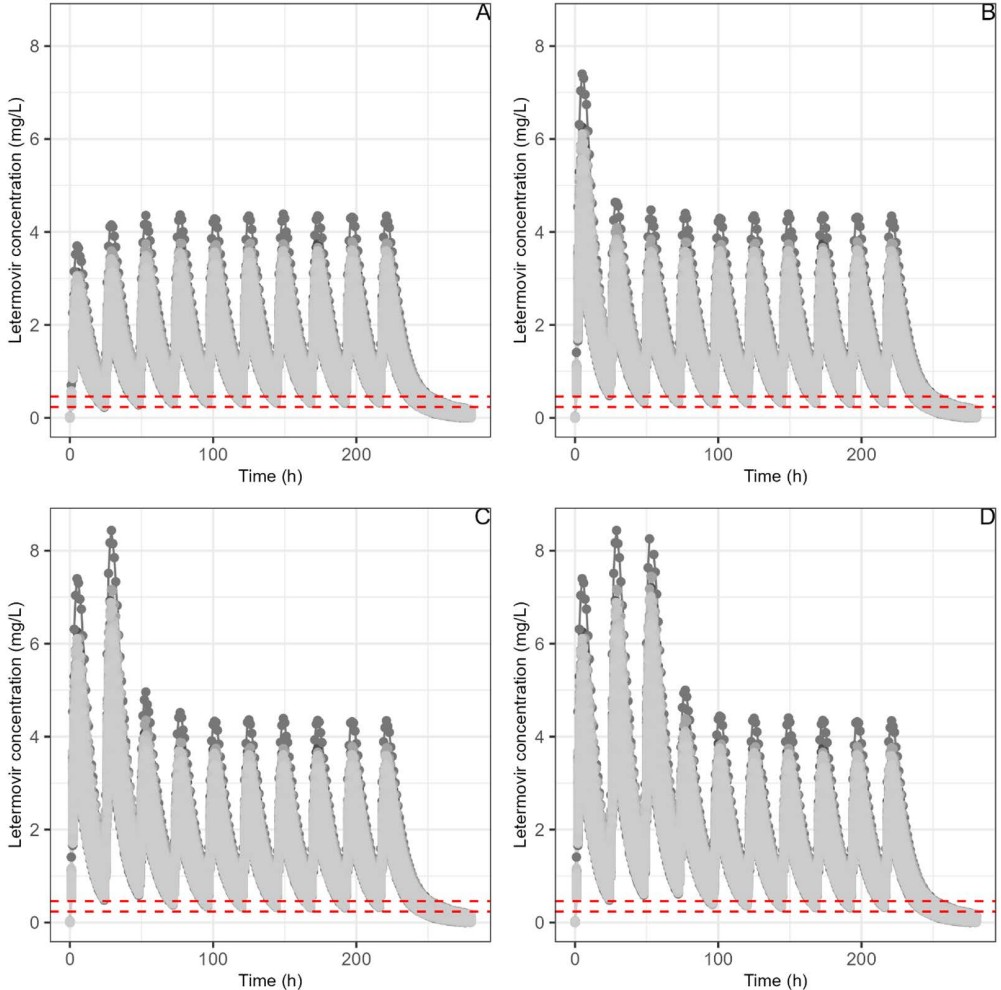

**Fig 2. Letermovir simulations for 1,000 patients for different loading doses schemes (after a typical 480 mg/daily dose, without CsA co-treatment).** Abbreviations: A: without loading dose; B: loading dose on day 1 of initiation treatment, C: loading dose on day 1- day 2 of initiation treatment, D: loading dose from day1 to day 3 1 of initiation treatment. The red dashed lines represent the 5*PA-IC50range.

### Missed dose and dosing regimen for resumption

For missed letermovir doses, simulations show median (IQR) trough plasma concentrations at 48 hours after the last dose to be 0.141 (0.116–0.169) mg/L and 0.382 (0.344–0.421) mg/L without and with cyclosporine, respectively. Fig 3 depicts the three scenarii considered for resuming treatment in 1,000 simulated patients.

When resuming treatment with the normal dosage (without cyclosporine), the median (IQR) concentrations 24 hours after restarting are: 0.478 (0.426–0.538) mg/L, 0.907 (0.806–1.018) mg/L, and 1.845 (1.655–2.054) mg/L for normal dose, double dose, and normal dose twice daily (BID) regimens, respectively. Similar trends are observed with cyclosporine co-administration (Fig 3). These simulated concentrations were higher than the target concentration ranges for PA-IC50 ([0.047–0.092] mg/L) and for 5*PA-IC50 ([0.235–0.460] mg/L).

For maribavir, the simulated median (IQR) of drug remaining at 24 hours after the last dose is 2.86 (1.32–5.83) mg/L. When simulating concentrations 12 hours after the recovery dose, the medians (IQR) are: 6.11 (3.96–9.17) mg/L for the standard dose, 8.38 (5.59–12.21) mg/L for 1.5 times the standard dose, and 10.62 (7.20–15.12) mg/L for double dose. To note, the target concentration ranges for PA-IC50 are [2.56–10.53] mg/L, with [11.29–52.67] mg/L as the target for

**Table 3. Results of the C12h/C8h (trough plasma concentration) and Cmax (maximal plasma concentration) simulations depending on different dosing regimen for maribavir.**

| | | 400mg BID | 400mg TID | 600mg BID | 800mg BID | 800mg TID |
|---|---|---|---|---|---|---|
| Trough concentration (mg/L) | Mean ±sd | 9.10±6.20 | 16.10±9.50 | 13.70±9.30 | 18.20±12.40 | 32.100±18.90 |
| | Geometric mean | 7.30 | 13.60 | 10.90 | 14.50 | 27.20 |
| | Median (IQR) | 7.80 | 14.00 | 11.50 | 15.30 | 27.90 |
| | | (4.90–11.90) | (9.40–20.00) | (7.10–17.60) | (9.50–23.40) | (18.80–40.30) |
| Cmax (mg/L) | Mean ± sd | 16.10±6.90 | 21.60±9.80 | 24.20±10.30 | 32.20±13.80 | 43.10±19.40 |
| | Geometric mean | 14.80 | 19.70 | 22.20 | 29.60 | 39.30 |
| | Median (IQR) | 14.90 | 19.70 | 22.30 | 29.60 | 39.20 |
| | | (11.30–19.70) | (14.80–26.40) | (16.90–29.50) | (22.50–39.20) | (29.50–52.60) |

Target concentrations for PA-IC50 are [2.56–10.53] mg/L and [11.29–52.67] mg/L for 5*PA-IC50

5*PA-IC50. Fig 4 shows the PTA in terms of PA-IC50 and 5*PA-IC50 ranges for different treatment resumption regimens. These PTAs were conducted at through concentration following the resumption of treatment after a missed dose.

## Discussion

In this study, we implemented models from the literature to evaluate, using Monte Carlo simulations, the impact of different dosing regimens on effective plasma concentrations. Regarding the choice of POPPK models for simulations, the 2 models used in this study were chosen because they were built based on the highest number of patients, hence increasing their robustness. Additionally, due to the novelty of these antiviral compounds, there are relatively few POPPK studies. To our knowledge, there is only one other physiological-based pharmacokinetic model available [23] for letermovir and for maribavir [24].

Simulations of virtual patients may yield unrealistic pharmacokinetic parameter combinations and profiles, especially if implemented without access to the variance-covariance matrix (a rare inclusion in literature). To address this issue, we applied filters to remove extreme values. Since the choice of a threshold for filtering is subjective, we explored four different quantile values to conduct a sensitivity analysis. As the quantile, influence was not clinically different between drugs, [1–99] interval was choosen, allowing maintaining the largest number of simulations. Overall, the median simulated trough plasma concentration values closely matched those reported in the literature, providing validation for our implementation. More precisely, for letermovir, previously reported C24h levels range from 0.259 mg/L to 1.20 mg/L, depending on co-treatments and administration routes. Prohn et al. [15] reported similar findings in their simulations, with median (min-max) C24h of 0.50 (0.20–1.80) mg/L and 1.20 (0.25–3.50) mg/L without and with cyclosporine, respectively, administered orally. Royston et al. [25] showed a median (IQR) of 0.980 (0.455–1.72) mg/L and a mean±SD of 1.123±0.875 mg/L with cyclosporine. Without cyclosporine, they reported a median (IQR) of 0.259 (0.119–0.542) mg/L and a mean±SD of 0.664±0.0013 mg/L. For maribavir, previously reported C24h levels after a 400mg BID dosage range from 4.9 to 7.0 mg/L, based on both *in vitro* and *in vivo* analyses. Specifically, Sun et al. [19] reported a simulated median (IQR) C24h of 7.0 (3.0–13.0) mg/L, while Papanicolaou et al. [20] observed a median (range) of 5.5 (1.8–18.9) mg/L. The drug'sSPC [7] lists a geometric mean (CV%) of 4.9 (89.7%) mg/L.CMV can infect a wide range of cell types, including endothelial, epithelial, fibroblastic, dendritic, nerve, and smooth muscle cells, as well as macrophages and hepatocytes. This broad tropism enables CMV to infect multiple tissues and be detected in most organs. Letermovir and maribavir are both highly protein-bound (98.2% and 98% respectively [5,7]) and have fairly low volumes of distribution (45L and 27L respectively [5,7]), indicating limited distribution into deep compartments. Therefore, administering a loading dose or higher doses may be advantageous to quickly saturate plasma compartments and enhance distribution into deeper compartments.

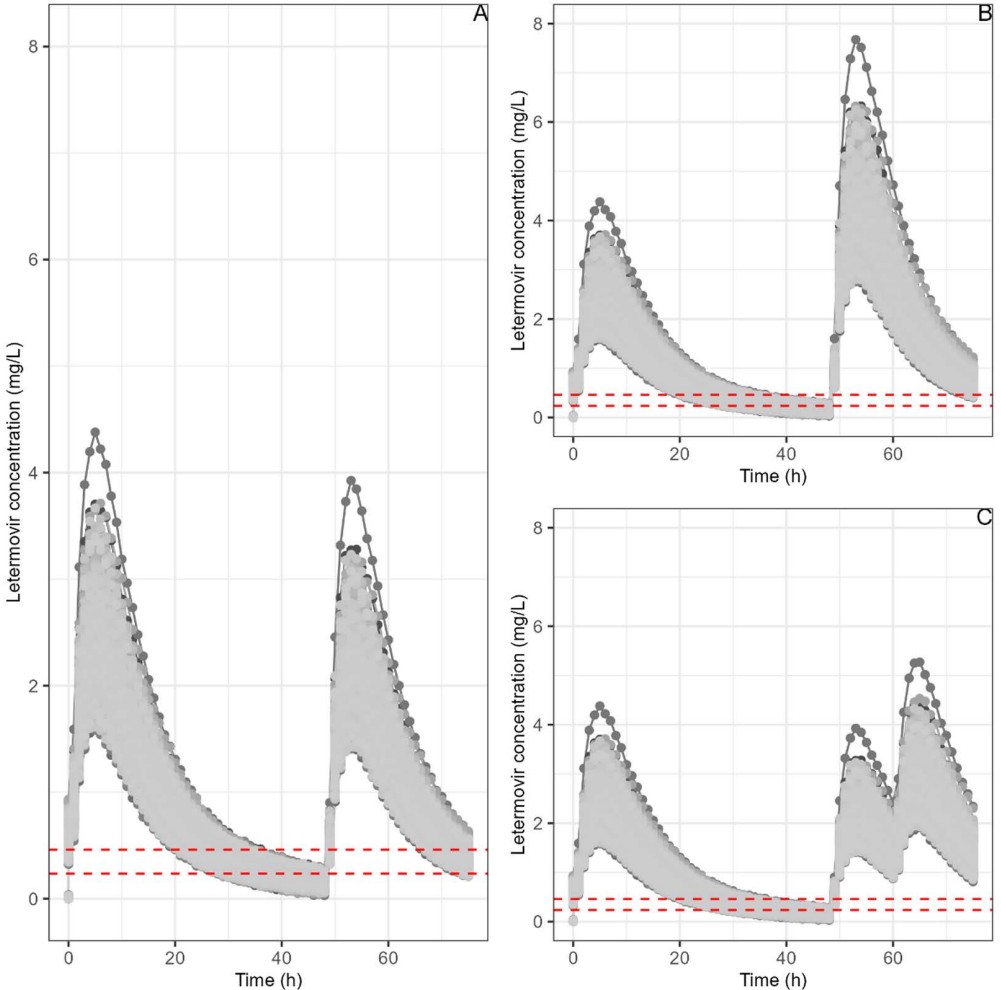

**Fig 3. Letermovir simulations for 1,000 patients for 3 different dosing resumption regimens (after a typical 480 mg/day dose and without cyclosporine co-treatment).** Abbreviations: **A:** Dosing resumption at normal posology; **B:** Dosing resumption with a double dose; **C:** Dosing resumption with normal posology BID. The red dashed lines represent the 5*PA-IC50range.

Regarding the results for letermovir, we observed an SS achieved between as early as day 4 and as late as day 6, depending on the route of administration and co-treatment, whereas the drug's SPC [5] reports an SS between days 9 and 10. These simulations suggest a faster stability of pharmacokinetic profiles compared to the SPC, justifying the interest of a loading dose to rapidly saturate cells with effective concentrations. Additionally, the simulations indicate that despite halving the daily dosage from 480mg to 240 mg, the combination with CsA doubles letermovir trough plasma concentrations.

Regarding the study of dosing regimens with loading doses, it appears that a single loading dose of letermovir on the first day of treatment is enough to reach effective plasma concentrations. Fig 1A illustrates that without a loading dose, higher ranges of protein-adjusted 50% inhibitory concentrations have lower chances of success, whereas a single loading dose ensures 100% success upon first administration. Given the drug's mechanism of action and the need to achieve effective plasma concentrations rapidly, this supports the implementation of a loading dose at treatment initiation. It is important to note that when letermovir is administered without CsA, despite halving the dosage, the success rates remain

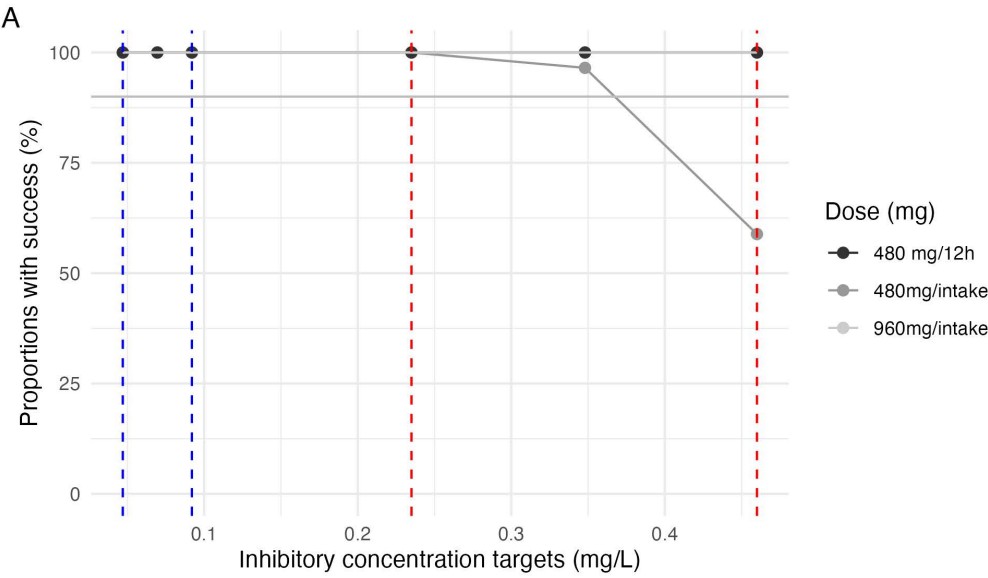

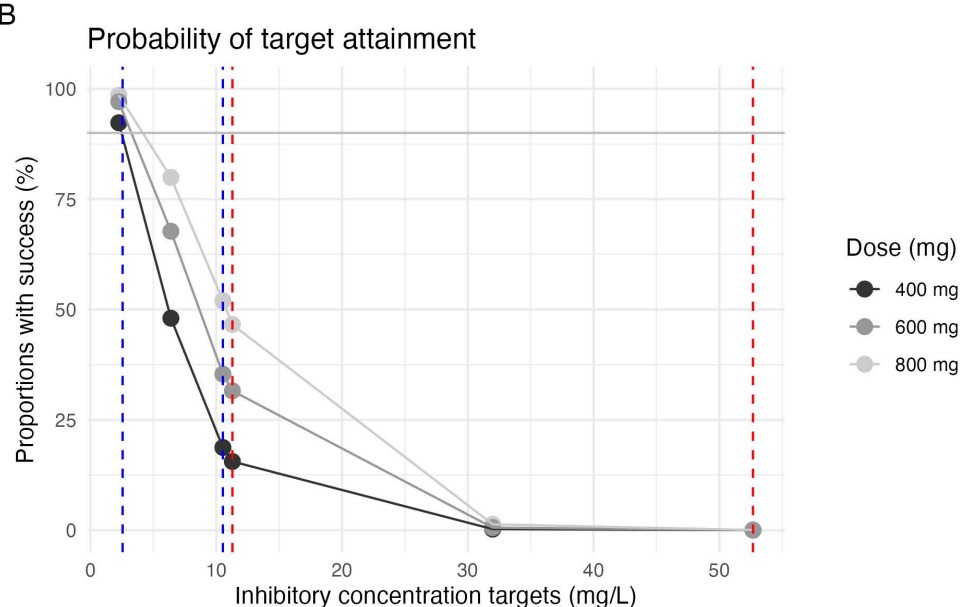

**Fig 4. PTA for different trough concentration higher than inhibitory concentration or 5 times PA-IC50 after different treatment resumption scenarii following a missed dose for letermovir (A) and maribavir (B).** The blue dashed lines represent the IC50 ranges, specifically [0.047–0.092] mg/L for letermovir and [2.560–10.530] mg/L for maribavir. The red dashed lines represent the 5*PA-IC50 ranges, specifically [0.235–0.460] mg/L for letermovir and [11.290–52.670] mg/L for maribavir. The grey solid line represents the proportion of 90% success.

at 100% even for the highest ranges of 5*PA-IC50, even without a loading dose. Safety data supports this approach. A phase 1 study [6] evaluated letermovir tolerability at doses ranging from 120mg to 960mg per dose. No significant increase in side effects was observed with higher doses. However, this study has limitations: a small group (n = 9) of healthy women were included, which doesn't represent the real-world patient population.

To build on these findings, a phase 2 study [17] explored the efficacy and safety of 60mg, 120mg, and 240mg doses. Higher doses were not tested, and the 240mg dose demonstrated the best balance between effectiveness and tolerability.

Informed by these earlier trials, a phase 3 study [9] confirmed the drug's efficacy and safety at 480mg without CsA and 240mg with CsA. Therefore, higher letermovir dosages were not investigated further, particularly regarding their impact on achieving effective concentrations or potential resistance development. The rapid emergence of resistance in association to underdosage or missing doses was recently underlined in some case reports for letermovir [26]. The high association between maribavir resistance and persisting or recurrent viral DNA replication was also noticed [27]. Considering both drugs, their direct mechanism of action, and their lower genetic barrier could justify to rapidly reach and maintain a trough concentration largely above their respective IC50, for prophylaxis or treatment.

The recommended treatment resumption strategy after a missed letermovir dose, as outlined in the SPC [5], may not be the most effective. Resuming with a normal dose following a 24-hour missed dose leads to lower median C24h levels compared to typical values (0.141mg/L vs 0.591mg/L and 0.382mg/L vs 1.28mg/L, without and with CsA, respectively). Conversely, taking a normal dose every 12 hours after a missed dose results in significantly higher median simulated plasma concentrations than usual, potentially increasing the risk of side effects (1.845mg/L vs 0.591mg/L and 3.289mg/L vs 1.280mg/L, without and with CsA, respectively). Based on these simulations, doubling the dose after a missed dose appears to be the most optimal strategy (0.907mg/L vs 0.591mg/L and 1.873mg/L vs 1.280mg/L, without and with CsA, respectively). In the absence of co-administration with CsA, Fig 4A illustrates that the current recommended dosage of resuming at the normal dose does not maintain a chance of success for the higher range of PA-IC50. Conversely, both doubling the dosage and administering normally twice daily achieve 100% success. It is also noteworthy that co-treatment with CsA ensures 100% success despite halving the dosage. Since the 480mg and 240mg tablets are not divisible, we did not explore dosages in increments of 0.5 times the dose, as this would be impractical for patients in real-world settings. Finally, the scenario of skipping two doses was not investigated due to its low likelihood in clinical practice. Indeed, "forgiveness strategies" sometimes used for antiretrovirals in HIV treatment [26–28] are not yet studied or standardized for CMV management. Thus, we did not include them in our study, but they could be explored in future simulations if deemed appropriate by the scientific community.

Maribavir's safety profile has been established up to 2400mg/day, allowing for exploration of higher dose regimens [8,29]. The current standard dosage of 400mg BID was chosen because the 100mg BID dose proved ineffective in controlling CMV disease [30]. Studies by Papanicolaou et al. and Maertens et al. [20,31] suggest that 800mg and 1200mg BID dosages are no more likely to cause side effects compared to 400mg BID, adverse effects were responsible for only three treatment discontinuations out of 120 patients [20]. Additionally, higher doses sligthly reduced the median time to achieve viral clearance (22 days for 1200mg BID vs 24 days for 400mg BID). Maribavir's marketing authorization was based on the lower 400mg BID dosage due to its effectiveness and similar viral clearance rates across all studied doses, although the SOLSTICE study [10] shows relatively modest viral clearance rates at this dosage. However, a limitation of this choice is the small sample size (n = 40) in each dosing group, which may have obscured a dose-response relationship. Given the rise in anti-CMV resistance and the need for rapid viral suppression, these simulations suggest that a higher reference dosage could be considered. The simulations for the 800mg BID regimen align well with the findings from the phase 2 observational study by Papanicolaou et al. [20]. Their reported median (IQR) for C12h and Cmax were 10.8 (1.5–21.9) mg/L and 25.0 (13.1–41.3) mg/L, respectively, while our simulations show a median (IQR) of 15.3 (9.5–23.4) mg/L and 29.6 (22.5–39.2) mg/L for C12h and Cmax, respectively. It's important to note that dosing regimens of 800mg TID was not, to our knowledge, explored during maribavir's clinical development. The 400mg TID and 600mg BID doses have only been studied in phase I trial [32]. This limits our ability to compare them with observational data. If we use the PA-IC50 range as comparative values, Fig 1B and 1C show that the 400mg BID regimen only covers the lower end of the PA-IC50 range. Notably, only the 800mg every 8 hours regimen achieves a 90% probability of success across the entire PA-IC50 range and the lower end of the 5*PA-IC50 range. However, this very high dosage increases the rate of dysgueusia. Moreover, treatment costs would be significantly impacted, with approximately $10,000 per box of 28 tablets.

The SPC [7] recommends taking the standard dose after a missed dose of maribavir. However, simulations suggest this approach results in slightly lower median C12h (6.11 mg/L) compared to steady state levels (7.8 mg/L). Doubling the dose, as advised against by the SPC [7], is also not ideal, as simulations predict significantly higher median concentrations (10.62 mg/L). A more balanced approach appears to be taking 1.5 times the standard dose, which achieves median concentrations closer to SS (8.38 mg/L vs. 7.8 mg/L). Fig 4B demonstrates that, in terms of PTA for the PA-IC50 range, any dosage used only covers the lowest part of the PA-IC50 spectrum. However, the dose recommended by the MA exhibits the lowest chances of success, suggesting a potential need to modify the recommendation to resume the same dose after a missed dose.

This work has some limitations. Firstly, the indication for letermovir is primarily for the prophylaxis of CMV infection and disease in adult CMV-seropositive recipients of an allogeneic HSCT, recently extended to kidney transplant recipients. This drug has been tested in a single curative trial involving only 10 patients. Therefore, the significance of a loading dose in prophylaxis can be questioned. Nevertheless, considering the most severe context, where patients are CMV+, the prophylactic treatment should be as effective as possible. Additionally, letermovir is sometimes used off-label, such as in France for refractory infections, with a national registry established under the auspices of the French Society for Infectious Pathology (SPILF) and the National Reference Center for Herpesviruses. The need for quickly effective dosing is thus understandable. Moreover, it has been shown that maribavir exposure is significantly impacted by food intake, with fasting reducing through and peak concentrations by approximately 30% [8]. Since the POPPK models used do not account for this covariate, we were unable to study the impact of this phenomenon in our simulations. Developing a model that includes this important covariate could be a promising area for future research. Finally, we chose to study only dose increases rather than the synergistic effects of anti-CMV combination therapies. Monte Carlo simulations using POPPK models typically only allow for the study of monotherapy, unless a covariate is directly indicated in the model, which is not the case here.

## Conclusions

In conclusion, the emergence of resistance to traditional anti-CMV treatments highlights the necessity for new therapeutic options, with letermovir and maribavir showing promise due to their different pharmacokinetic, distinct mechanisms of action and dosing considerations. Monte Carlo simulations, utilizing population pharmacokinetic models, have provided insights into various dosing regimens, suggesting that loading doses for letermovir could hasten the attainment of stable therapeutic concentrations, while higher doses of maribavir might facilitate quicker viral clearance. Additionally, our study evaluated missed doses for both drugs, indicating that a double dose for letermovir and 1.5 times the dose for maribavir are optimal for resuming treatment to maintain effective plasma concentrations. These results have practical implications for medication adherence and treatment efficacy in CMV infection management, although further clinical validation is warranted to refine dosing recommendations for real-world application.

## Supporting information

**S1 Fig. Histogram of simulated letermovir C24h.**
(TIF)

**S2 Fig. Histogram of simulated maribavir C24h.**
(TIF)

**S1 Table. Simulated letermovir concentrations (µg/L) from days 1–10, without ciclosporin, for various loading dose.**
(PDF)

**S2 Table. Simulated letermovir concentrations (µg/L) from days 1–10, with ciclosporin, for various loading dose.**
(PDF)

**S1 File. R code used for the letermovir analysis, presented as an HTML-knitted R Markdown report.**
(HTML)

**S2 File. R code used for the maribavir analysis, presented as an HTML-knitted R Markdown report.**
(HTML)

## Author contributions

**Conceptualization:** Yeleen Fromage, Hamza Sayadi, Sophie Alain, Jean-Baptiste Woillard.

**Formal analysis:** Yeleen Fromage, Jean-Baptiste Woillard.

**Investigation:** Sophie Alain, Pierre Marquet, Gilles Peytavin, Jean-Baptiste Woillard.

**Methodology:** Yeleen Fromage, Hamza Sayadi, Jean-Baptiste Woillard.

**Software:** Yeleen Fromage, Hamza Sayadi, Jean-Baptiste Woillard.

**Supervision:** Sophie Alain, Pierre Marquet, Gilles Peytavin, Jean-Baptiste Woillard.

**Validation:** Yeleen Fromage, Hamza Sayadi, Gilles Peytavin, Jean-Baptiste Woillard.

**Visualization:** Yeleen Fromage, Hamza Sayadi, Jean-Baptiste Woillard.

**Writing – original draft:** Yeleen Fromage, Jean-Baptiste Woillard.

**Writing – review & editing:** Sophie Alain, Pierre Marquet, Gilles Peytavin.

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
