## [Decision Letter · Decision Letter 0]

10 Jan 2025

PONE-D-24-45733Optimizing CMV therapy: population pharmacokinetics and Monte Carlo simulations for letermovir and maribavir dosagePLOS ONE

Dear Dr. Woillard,

Thank you for submitting your manuscript to PLOS ONE. After careful consideration, we feel that it has merit but does not fully meet PLOS ONE’s publication criteria as it currently stands. Therefore, we invite you to submit a revised version of the manuscript that addresses the points raised during the review process.

We look forward to receiving your revised manuscript.

Kind regards,

Hideo Kato

Academic Editor

PLOS ONE

**Journal Requirements:**

SA received research grants or contracts paid to institutions, travel grants, advisory board fees and speaker fees from MSD and Takeda, GP received travel grants and speaker fees from MSD and Takeda and PM received speaker fees from Takeda  

We note that one or more of the authors are employed by a commercial company. 

“The funder provided support in the form of salaries for authors, but did not have any additional role in the study design, data collection and analysis, decision to publish, or preparation of the manuscript. The specific roles of these authors are articulated in the ‘author contributions’ section.”

Reviewers' comments:

Reviewer's Responses to Questions

**Comments to the Author**

1. Is the manuscript technically sound, and do the data support the conclusions?

Reviewer #1: Yes

2. Has the statistical analysis been performed appropriately and rigorously? 

Reviewer #1: Yes

3. Have the authors made all data underlying the findings in their manuscript fully available?

Reviewer #1: Yes

4. Is the manuscript presented in an intelligible fashion and written in standard English?

Reviewer #1: Yes

5. Review Comments to the Author

**Reviewer #1: ** This manuscript provided invaluable simulations on optimal pharmacokinetics for Letermovir and Maribavir when dosing for CMV prophylaxis. It provided additional information on loading dose and on the coping strategy on missing dose for those candidates receiving CMV prophylaxis. However, there are still several aspects that need to be elucidated clearly.

First, even a study of simulation from previous reports, it was still suggested to have the IRB approval in the case of human studies.

Second, it was suggested to recheck the wording being consistent and being adequately presented. There are some mistakes disclosed in text when using mg/L but mis-labeled μg/L, such as the Letermovir reported IC50 values in the literature range from 0.00086 μg/L to 0.00166 mg/L in line 86-87 in page 5, the target concentration intervals for PA-IC50 are [2.56-10.53] 171 μg/L, and [11.29-52.67] mg/L for 5*PA-IC50 in line 170-171 in page 9, etc.; and the error number presented as median (IQR) C24h of 567.8 (0.508-0.638) mg/L in line 201-202 in page 10. It was also found discrepancy in wordings such as 5*PA-IC50, 5PA-50, and 5*IC50 in text and footnote of tables. We could not find the full name of PA in text, but the full name of PTA. Please recheck above conditions for twice before resubmission. Furthermore, we could not find the information of Letermovir IC50 in reference 5. Please cite adequate reference with corresponding information.

Third, please remove all the reference citations in result part. The findings of results should be simple and clear based on your esteem study, and should never mix with results of previous studies. To have a better presentation, please move those comparison to the discussion part.

Fourth, we could not find the full name of IIV in line 127, page 7. For it appears for once, please spell the full name only. Furthermore, there were no any IIV in Table 1, and it was suggested to remove from the footnote.

Final, the resolution of figures was not friendly to read. Please recheck and upload those figures with better resolution.

6. PLOS authors have the option to publish the peer review history of their article (what does this mean? ). If published, this will include your full peer review and any attached files.

**Do you want your identity to be public for this peer review?** For information about this choice, including consent withdrawal, please see our Privacy Policy .

Reviewer #1: **Yes: ** Liang-Yu Chen

---

## [Author Response · Author response to Decision Letter 0]

3 Feb 2025

Reviewer #1

1. First, even a study of simulation from previous reports, it was still suggested to have the IRB approval in the case of human studies.

Answer: Following your comments, we confirm that a favorable opinion from the ethics committee was obtained (IRB approval: 03-2025-01), and this information has now been included in the manuscript (approval letter in supplemental data).

2. Second, it was suggested to recheck the wording being consistent and being adequately presented. There are some mistakes disclosed in text when using mg/L but mis-labeled μg/L, such as the Letermovir reported IC50 values in the literature range from 0.00086 μg/L to 0.00166 mg/L in line 86-87 in page 5, the target concentration intervals for PA-IC50 are [2.56-10.53] 171 μg/L, and [11.29-52.67] mg/L for 5*PA-IC50 in line 170-171 in page 9, etc.; and the error number presented as median (IQR) C24h of 567.8 (0.508-0.638) mg/L in line 201-202 in page 10. It was also found discrepancy in wordings such as 5*PA-IC50, 5PA-50, and 5*IC50 in text and footnote of tables. We could not find the full name of PA in text, but the full name of PTA. Please recheck above conditions for twice before resubmission. Furthermore, we could not find the information of Letermovir IC50 in reference 5. Please cite adequate reference with corresponding information.

Answer: Thank for these comments, which highlight typographical errors whose correction helps to avoid any misunderstanding. All numerical values have been harmonized to mg/L. Similarly, the acronym PA-IC50 is now the only one used throughout the text, and its full name has been explicitly provided upon first mention. The distinction between in vitro IC50 values obtained from a collection of clinical CMV isolates and the Pharmacologically Active IC50, which accounts for plasma protein binding, is now explicitly stated.

Regarding the letermovir’s IC50 value, we greatly appreciate the reviewer’s attention. The reference has been updated accordingly. We would like to note that, both in the Summary of product characteristics and in the newly added reference, the IC50 values are expressed in nM. For the sake of readability and simplification, we have converted these values into mg/L to maintain consistency in the units used throughout the manuscript. Thus, the IC50 values of 2.2 ± 0.7 nM correspond to the values mentioned in the text, ranging from 0.00086 mg/L to 0.00166 mg/L.

3. Third, please remove all the reference citations in result part. The findings of results should be simple and clear based on your esteem study, and should never mix with results of previous studies. To have a better presentation, please move those comparison to the discussion part.

Answer: We have made the suggested modifications, and the Results section now focuses solely on the findings from our simulations. Comparisons with previous studies have been moved to the Discussion section.

4. Fourth, we could not find the full name of IIV in line 127, page 7. For it appears for once, please spell the full name only. Furthermore, there were no any IIV in Table 1, and it was suggested to remove from the footnote.

Answer: The corrections have been made in the legend of Table 1 as well as in the main text.

5. Final, the resolution of figures was not friendly to read. Please recheck and upload those figures with better resolution.

Answer : Locally, the figures appear to be of high quality; however, it seems that the uploading process on the PLOS ONE platform may have affected their resolution. We will carefully ensure that the final publication version maintains high-quality figures to enhance readability

---

## [Editor Report · Decision Letter 1]

3 Mar 2025

Optimizing CMV therapy: population pharmacokinetics and Monte Carlo simulations for letermovir and maribavir dosage

PONE-D-24-45733R1

Dear Dr. Woillard,

We’re pleased to inform you that your manuscript has been judged scientifically suitable for publication and will be formally accepted for publication once it meets all outstanding technical requirements.

Kind regards,

Hideo Kato

Academic Editor

PLOS ONE
---

## [Editor Report · Acceptance letter]

PONE-D-24-45733R1

PLOS ONE

Dear Dr. Woillard,

I'm pleased to inform you that your manuscript has been deemed suitable for publication in PLOS ONE. Congratulations! Your manuscript is now being handed over to our production team.

Kind regards,

on behalf of

Dr. Hideo Kato

Academic Editor

PLOS ONE